# Regulated Ire1-dependent mRNA decay requires no-go mRNA degradation to maintain endoplasmic reticulum homeostasis in *S. pombe*

Nicholas R Guydosh[1,2†], Philipp Kimmig[3,4†], Peter Walter[3‡*], Rachel Green[1‡*]

[1]Department of Molecular Biology and Genetics, Howard Hughes Medical Institute, Johns Hopkins School of Medicine, Baltimore, United States; [2]Laboratory of Biochemistry and Genetics, National Institute of Diabetes and Digestive and Kidney Diseases, National Institutes of Health, Bethesda, United States; [3]Department of Biochemistry and Biophysics, Howard Hughes Medical Institute, University of California, San Francisco, San Francisco, United States; [4]Institute of Biochemistry, ETH Zurich, Zurich, Switzerland

**Abstract** The unfolded protein response (UPR) monitors and adjusts the protein folding capacity of the endoplasmic reticulum (ER). In *S. pombe*, the ER membrane-resident kinase/endoribonuclease Ire1 utilizes a mechanism of selective degradation of ER-bound mRNAs (RIDD) to maintain homeostasis. We used a genetic screen to identify factors critical to the Ire1-mediated UPR and found several proteins, Dom34, Hbs1 and Ski complex subunits, previously implicated in ribosome rescue and mRNA no-go-decay (NGD). Ribosome profiling in ER-stressed cells lacking these factors revealed that Ire1-mediated cleavage of ER-associated mRNAs results in ribosome stalling and mRNA degradation. Stalled ribosomes iteratively served as a ruler to template precise, regularly spaced upstream mRNA cleavage events. This clear signature uncovered hundreds of novel target mRNAs. Our results reveal that the UPR in *S. pombe* executes RIDD in an intricate interplay between Ire1, translation, and the NGD pathway, and establish a critical role for NGD in maintaining ER homeostasis.
DOI: https://doi.org/10.7554/eLife.29216.001

**\*For correspondence:**
peter@walterlab.ucsf.edu (PW);
ragreen@jhmi.edu (RG)

[†]These authors contributed equally to this work
[‡]These authors also contributed equally to this work

## Introduction

Membrane and secreted proteins fold and mature within the endoplasmic reticulum (ER) before they are delivered to other compartments in the secretory pathway or to the plasma membrane. An imbalance between the protein folding load and the protein folding capacity in the ER leads to an accumulation of unfolded or misfolded proteins, a condition referred to as 'ER stress.' ER stress triggers the unfolded protein response (UPR), a network of signal transduction pathways that drive transcriptional programs to expand the ER's protein folding capacity and reduce protein influx into the organelle through translational and mRNA degradative mechanisms, thereby ensuring that the organelle remains in or returns to homeostasis (*Walter and Ron, 2011*).

In metazoans, the UPR is orchestrated by three ER-resident sensors/signal transducers: the membrane tethered transcription factor ATF6 and the transmembrane kinases PERK and IRE1 ('Ire1' in accordance with the yeast nomenclature) (*Gardner et al., 2013*). Each sensor activates a downstream transcriptional gene expression program of UPR target genes. In addition to the transcriptional response, the PERK branch induces cell-wide attenuation of translation by phosphorylating the general eukaryotic translation initiation factor 2 (eIF2), reducing the protein folding load of the ER.

**eLife digest** Most proteins need to fold into a specific shape in order to work properly. As such, cells have developed a number of ways to sense and respond to stressful conditions that cause their proteins to fold incorrectly. One place this happens is within a network of tubes inside the cell called the endoplasmic reticulum; this is where proteins that are destined for the cell surface or other compartments in the cell become folded. The endoplasmic reticulum has a limited capacity to fold proteins. When it is overwhelmed, the cell temporarily stops making the proteins that use up this capacity. This action makes up part of a larger set of responses collectively referred to as the "unfolded protein response".

During the unfolded protein response, the production of some proteins is turned off when an enzyme called Ire1 cuts the transcript molecules that contain the instructions to build these proteins. Cutting these transcripts, however, creates a problem: it interrupts the translation of the transcript by the ribosome, the molecular machine that reads the genetic code to build proteins. Usually, a ribosome only comes off of a transcript when it arrives at a specific stop signal. Yet, ribosomes that run to the ends of broken transcripts never reach this signal and instead have to be rescued. If left without rescue, these stalled ribosomes could never be used again for translation of other transcripts, and the cell would lose the ability to make more proteins.

Guydosh, Kimmig et al. searched for new genes in the yeast *Schizosaccharomyces pombe* that are involved in the part of the unfolded protein response that occurs after the actions of the Ire1 enzyme. This search revealed that cells missing so-called ribosome rescue proteins (namely Dom34 and Hbs1) grow slowly under conditions that cause proteins to fold incorrectly. Guydosh, Kimmig et al. then looked to see where on the transcripts the ribosomes stall and remain un-rescued in the absence of these ribosome rescue proteins. These sites corresponded to places that were cut by Ire1, the majority of which were previously unknown. Together these findings indicate that ribosome rescue is a key part of the unfolded protein response in *S. pombe* because it removes ribosomes that stall at the broken ends of transcript molecules cut by the Ire1 enzyme.

An efficient and well-controlled response to conditions that cause proteins to fold incorrectly is important for human health. Loss of this control can lead to disorders as diverse as atherosclerosis, cancer and neurological diseases. By revealing that the unfolded protein response uses the ribosome rescue pathway, the findings improve our understanding of these health conditions and may open the door to new research and treatments.

DOI: https://doi.org/10.7554/eLife.29216.002

IRE1 catalyzes signaling through the most phylogenetically conserved branch of the UPR. It is a bifunctional transmembrane kinase/endonuclease activated by ER stress. In *S. cerevisiae* (and metazoans), Ire1 catalyzes the unconventional splicing of the mRNA encoding the transcription factor Hac1 (XBP-1 in metazoans) that activates a comprehensive transcription program (*Acosta-Alvear et al., 2007*; *Calfon et al., 2002*; *Cox and Walter, 1996*; *Travers et al., 2000*). Additionally, as first demonstrated in *D. melanogaster*, Ire1 initiates a reaction termed regulated Ire1-dependent mRNA decay, or RIDD, the selective decay of ER-bound mRNAs, thereby reducing the load of protein entering the ER (*Hollien and Weissman, 2006*). Conceptually, the effects of this reaction resemble that of PERK's translational attenuation that reduces the ER's protein folding load. RIDD occurs in mammalian cells and plants, but not in *S. cerevisiae*, where the *HAC1* mRNA splicing reaction is the sole output of UPR signaling (*Niwa et al., 2005*). In striking contrast, in *S. pombe* RIDD is the sole output of Ire1. Transcriptional regulation and the otherwise conserved mRNA splicing reaction are entirely absent. *S. pombe* Ire1 has been shown to induce the decay of a few dozen mRNAs, cleaving them between the G and C residues of a short consensus sequence UGC (UG/C) (*Kimmig et al., 2012*). This motif is too low in information content to specify engagement of select mRNAs with Ire1, and thus other, still unidentified features need to contribute to bring appropriate mRNAs into juxtaposition and facilitate their engagement with Ire1.

During RIDD, mRNAs that are endonucleolytically severed by Ire1 rapidly decay through the combined actions of cellular exoribonucleases (XRNs) in the 5'→3' direction and the exosome with the associated Ski complex in the 3'→5' direction (*Hollien and Weissman, 2006*; *Kimmig et al., 2012*).

Interestingly, virtually all Ire1 cleavage sites in *S. pombe* mRNAs are located within the coding sequences (CDSs). This observation is surprising, as mRNAs nicked within their CDSs are bound to induce translational stalls as ribosomes encounter the 3' ends of the truncated mRNAs. This notion predicts that ribosomes stalled at the end of truncated mRNAs must be rescued by an active clearance mechanism, such as the ribosome rescue/mRNA decay pathway known as 'no-go decay' (NGD) (*Doma and Parker, 2006*; *Shoemaker and Green, 2012*; *Tsuboi et al., 2012*).

Recent biochemical and genome-wide ribosome foot-printing studies of NGD in *S. cerevisiae* showed that the Dom34/Hbs1 complex in cooperation with Rli1 promotes dissociation of stalled ribosomes on truncated mRNAs (a form of ribosome recycling termed 'rescue') (*Guydosh and Green, 2014*; *Pisareva et al., 2011*; *Shoemaker et al., 2010*). Additionally, the NGD pathway triggers endonucleolytic cleavage of the mRNA upstream of the stalled ribosomes carried out by a still unidentified endonuclease (which we here refer to as 'NGDase'), liberating ribosome-free mRNA fragments accessible to exonucleases (*Doma and Parker, 2006*; *Tsuboi et al., 2012*). NGD is critical for rescuing stalled ribosomes and therefore maintaining ribosome homeostasis and is connected to the degradation of incomplete protein products through ubiquitylation and proteasome digestion (*Bengtson and Joazeiro, 2010*; *Brandman et al., 2012*; *Shen et al., 2015*). In *S. cerevisiae*, NGD serves as an important quality control mechanism, responding to premature polyadenylation events in ORFs, as translation of the poly(A) tail stalls ribosomes and triggers subsequent decay (*Guydosh and Green, 2017*).

Here, we discovered that in *S. pombe* the NGD machinery Dom34/Hbs1 and the exosome-associated Ski-complex are critical players in the UPR, acting downstream of Ire1-catalyzed mRNA cleavage. Further, using short-read ribosome profiling methodology, we identified hundreds of novel mRNA targets of Ire1. The precise and widespread nature of these target sites allowed us to show that stalled ribosomes serve as a ruler to template regularly spaced upstream mRNA cleavage events. Our results reveal that the UPR in *S. pombe* executes RIDD in an intimate interplay between Ire1, translation, and the NGD surveillance pathway.

## Results

### A genetic screen reveals novel factors critical to RIDD in *S. pombe*

To identify additional genes involved in the UPR in *S. pombe*, we performed a quantitative, genome-wide screen for mutants resulting in altered fitness compared to wild type (WT) cells when grown on limiting concentrations of tunicamycin (Tm), a widely used UPR inducer that acts by inhibiting N-glycosylation in the ER lumen. To this end, we analyzed 2346 yeast strains deleted for non-essential genes by quantifying colony size differences in the absence or presence of Tm (*Figure 1A*). By using a z-score cut-off of ±2, we identified 180 gene deletions that showed a significant change in cell fitness by Tm treatment, including 76 gene deletions that suppressed Tm-induced growth defects (*Figure 1A*, above blue dotted line) and 104 gene deletions that sensitized cells to Tm (*Figure 1A*, below red dotted line; *Supplementary file 1*).

Gene ontology (GO) analysis of gene deletions suppressing Tm growth defects showed enrichment for genes encoding proteins involved in vesicle transport and located on the cell surface (*Figure 1B*). Consistent with this result, prior studies have shown that changes in cargo transport within the secretory pathway can cause this effect (*Liu and Chang, 2008*). GO analysis of Tm-sensitizing deletions identified an enrichment in genes encoding glycosylation enzymes and integral membrane proteins, as well as factors mediating mRNA catabolic processes (*Figure 1B*). The latter class included the NGD components Ski2, Ski7, Dom34 and Hbs1. Since the unfolded protein response in *S. pombe* relies exclusively on RIDD and could be affected by defects in ribosome rescue, we henceforth focused our investigation on the NGD factors to determine how their actions might synergize with the UPR.

To validate that the growth defects related to RIDD and to exclude the possibility of potential false positive hits resulting from suppressor mutations in the deletion library, we reconstructed deletions of each of the four genes and plated the mutant and control strains on Tm plates. As shown in *Figure 1C*, the growth defects in *S. pombe* cells harboring *hbs1*, *dom34*, *ski2*, and *ski7* deletions impaired growth on Tm plates as compared to WT cells. Importantly, plating assays with the corresponding deletions in the NGD pathway in *S. cerevisiae* failed to exhibit growth defects under ER

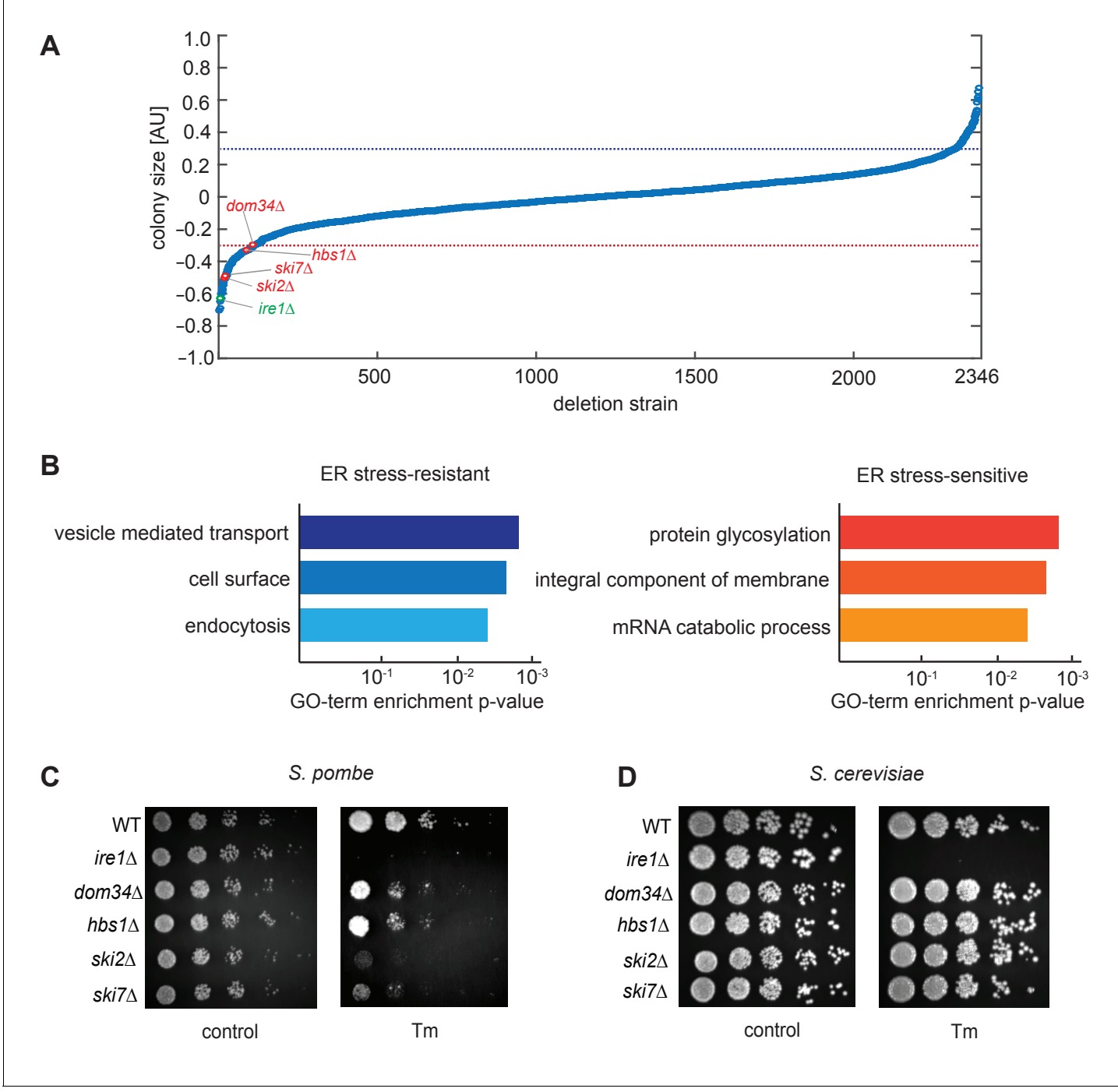

**Figure 1.** Genetic screen for factors involved in the unfolded protein response. (**A**) A chemical genetic screen of a non-essential deletion library. Each strain is plotted against the log ratio of colony size with and without addition of the ER stress inducer tunicamycin (Tm). Significant sensitive or resistant genes were identified by a standard-score (z-score) >2 (blue line) or <−2 (red line), corresponding to colony sizes measuring more than two standard deviations from the mean. Strains of interest missing no-go or nonstop decay factors (red) and the unfolded protein response factor *ire1* (green) are indicated. All tested strains are listed in ***Supplementary file 1***. (**B**) Gene-ontology (GO) analysis of enriched sensitive or resistant genes. (**C, D**) Viability assay by serial dilution of either *S. pombe* or *S. cerevisiae* wild type, *ire1Δ, dom34Δ, hbs1Δ, ski2Δ* and *ski7Δ* cells spotted on solid media with or without Tm (*S. pombe*: 0.03 μg/ml; *S. cerevisiae*: 0.3 μg/ml). Plates were photographed after 3 days of growth at 30°C. Note that although *ski2Δ* cells did not form colonies upon plating on Tm-containing media, they survived long enough after DTT addition in liquid media to allow for the footprinting analyses performed in this work.

DOI: https://doi.org/10.7554/eLife.29216.003

The following figure supplement is available for figure 1:

*Figure 1 continued on next page*

*Figure 1 continued*

**Figure supplement 1.** Northern blot analysis of total RNA extracted from wild type (WT) and *ski2Δ* mutant cells.

DOI: https://doi.org/10.7554/eLife.29216.004

stress conditions (*Figure 1D*). These results point to a central role for the NGD pathway in the UPR in *S. pombe.*

## Ribosome profiling reveals role for NGD in the *S. pombe* UPR

In light of a potential role for NGD in the *S. pombe* UPR, we asked whether stalled ribosomes found on Ire1-cleaved mRNAs would accumulate in NGD defective strains. To this end, we performed ribosome profiling in *S. pombe*, sequencing ribosome-protected mRNA fragments ('footprints') ranging from 15 to 34 nucleotides in length. In our experiment, we included WT, *dom34Δ*, *ski2Δ*, and *dom34Δ/ski2Δ* strains in the presence and absence of DTT, which induces the UPR by disrupting disulfide bond formation in the ER lumen. Since *ski2Δ S. pombe* cells show a severe growth defect upon Tm treatment (e.g., see *Figure 1C*), we performed ribosome foot-printing after a short exposure of DTT (60 min) to catch early events and minimize pleiotropic effects. We confirmed that at this early time point the Ire1 endonuclease was active by monitoring the accumulation of the cleavage product of *gas2* mRNA as a reporter RIDD target (*Figure 1—figure supplement 1*). As is customary in ribosome profiling experiments, we controlled for changes in mRNA abundance by simultaneously performing mRNA-Seq on the same samples.

Consistent with previous observations of ribosome footprint size distribution in *S. cerevisiae* (*Ingolia et al., 2009*), we found in the *dom34Δ/ski2Δ* strain that most of the ribosome footprints (~75%) were in the canonical range of 28–31 nucleotides (*Figure 2A*). We also observed smaller populations of footprints either 20–22 nts in length (~5% of the population; corresponding to ribosomes predicted to be in an alternative state of the translation cycle [*Lareau et al., 2014*]) or 15–18 nts in length (~10% of the population; predicted to be stalled on truncated mRNA ends [*Guydosh and Green, 2014*]). As we further elaborate below, the population of footprints also reports on the formation of 'disomes' (stacked ribosomes in direct contact with each other) because RNase 1, which is used to generate protected ribosome footprints, can only partially digest the mRNA between two closely-stacked ribosomes, thus yielding a larger footprint size. For reference, we also computed the size distribution for a WT strain (*Figure 2—figure supplement 1*).

We next compared the change in mRNA levels ± UPR induction in WT cells to the change in short footprint (15–18 nt) density ±UPR induction in *dom34Δ/ski2Δ* cells (*Figure 2B*). The rationale of this experiment was based on the prediction that in WT cells, UPR induction leads to degradation of the Ire1-generated fragments, whereas in *dom34Δ/ski2Δ* cells the 5' cleavage products are stabilized (due to the *ski2* knockout) and ribosomes are stabilized at their 3' ends to yield short footprints (due to the *dom34* knockout). Indeed, we found that a large fraction of the mRNAs that were degraded upon UPR induction in WT cells (*Figure 2B*, points left of the center cloud) were enriched in short footprints in *dom34Δ/ski2Δ* cells (*Figure 2B*, upper left quadrant). Importantly, this group of mRNAs includes the majority of RIDD targets previously identified by mRNA-Seq (*Figure 2B*, red dots) (*Kimmig et al., 2012*), thus validating the assumptions of our experimental strategy.

In a similar analysis, we further asked whether mRNAs that are enriched in short footprints after UPR induction correspond to those that are stabilized when Ire1 is deleted (*Figure 2C*, upper left quadrant). Again, many of these mRNAs corresponded to the previously identified RIDD transcripts (*Figure 2B* right, red dots) (*Kimmig et al., 2012*). Taken together, these data establish that the accumulation of short ribosome footprints upon UPR induction is a signature of RIDD, likely because these footprints are derived from ribosomes stalled at cleavage sites generated by Ire1.

## Peaks on individual transcripts reveal Ire1 cleavage sites

To identify the specific sites of ribosome stalling, we next examined the read distribution across genes that were enriched in short footprints upon UPR induction (*Figure 3*). In these examples, and in all profiling data shown in this work, we assigned the counts of mapped ribosome footprints according to their 3' ends. This method was used for long (25–34 nt) reads because our previous studies suggest that RNase 1 trims the 3' end of the footprint more precisely than the 5' end

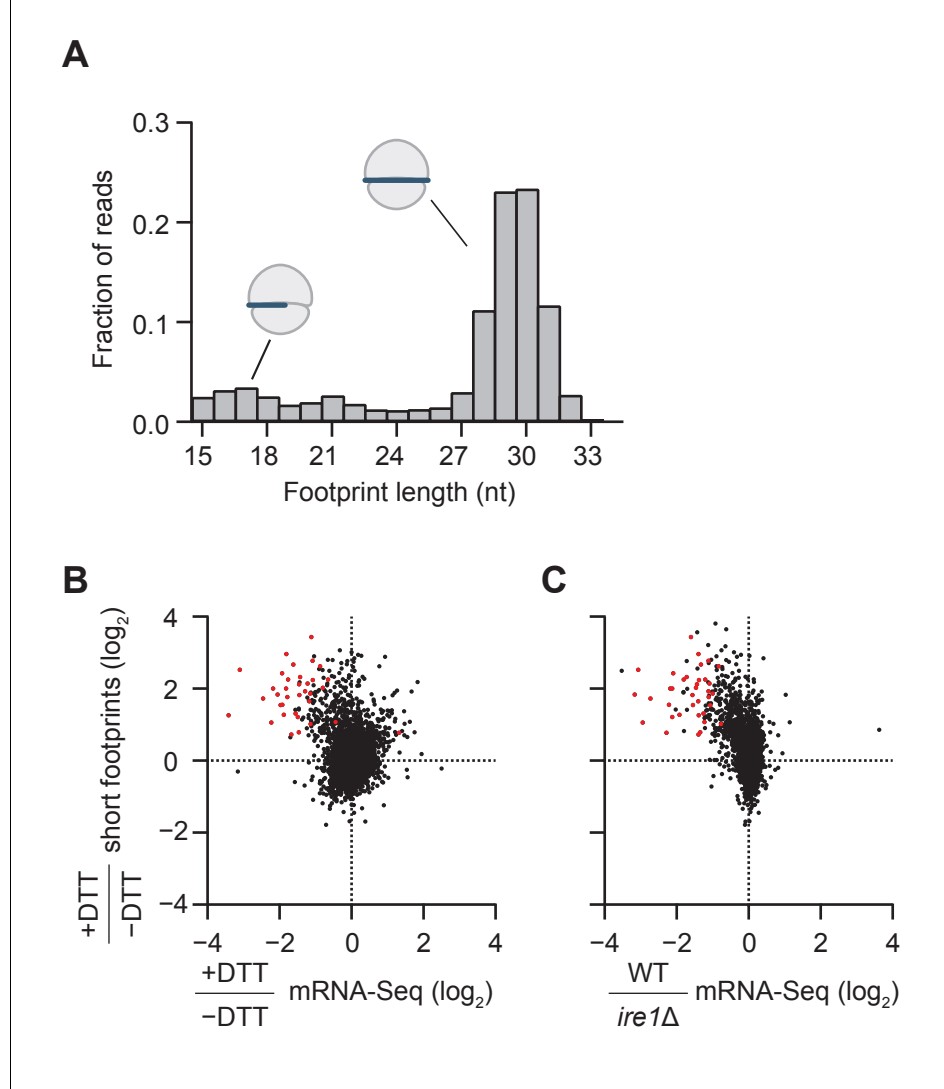

**Figure 2.** Short ribosome footprints are enriched on RIDD targets. (**A**) Size distribution of ribosome footprints mapped without mismatches to the transcriptome for *dom34Δ*/*ski2Δ* in the presence of DTT. Short footprints on truncated mRNAs correspond to 15–18 nt reads (*Guydosh and Green, 2014*) and ribosomes with a fully-occupied mRNA channel correspond to 28–31 nt (*Ingolia et al., 2009*). The small peak at 21 nt corresponds to ribosomes in an alternate conformation (*Lareau et al., 2014*). (**B**) Comparison of gene enrichment ratio (+DTT/−DTT) for 15–18 nt footprint data in the *dom34Δ*/*ski2Δ* background (y-axis) and mRNA-Seq data (x-axis) under the same growth conditions in the WT background. (**C**) Comparison of the same short footprint data ratio (y-axis) against mRNA-Seq data (x-axis) for WT/*ire1Δ* under UPR induction. Annotation of 39 previously-identified RIDD mRNA targets (red dots in B and C) and WT/*ire1Δ* mRNA-Seq ratios (**C**) are from (*Kimmig et al., 2012*). Anti-correlated trends in B and C show that mRNAs degraded by RIDD (low mRNA-Seq ratios when DTT added or *ire1* deleted) are enriched in stalled ribosomes (footprints), consistent with the prediction that cleavage sites created by Ire1 stall ribosomes on short (15–18 nt) footprints.

DOI: https://doi.org/10.7554/eLife.29216.005

The following figure supplement is available for figure 2:

**Figure supplement 1.** Size distribution of ribosome footprints as in Fig.

DOI: https://doi.org/10.7554/eLife.29216.006

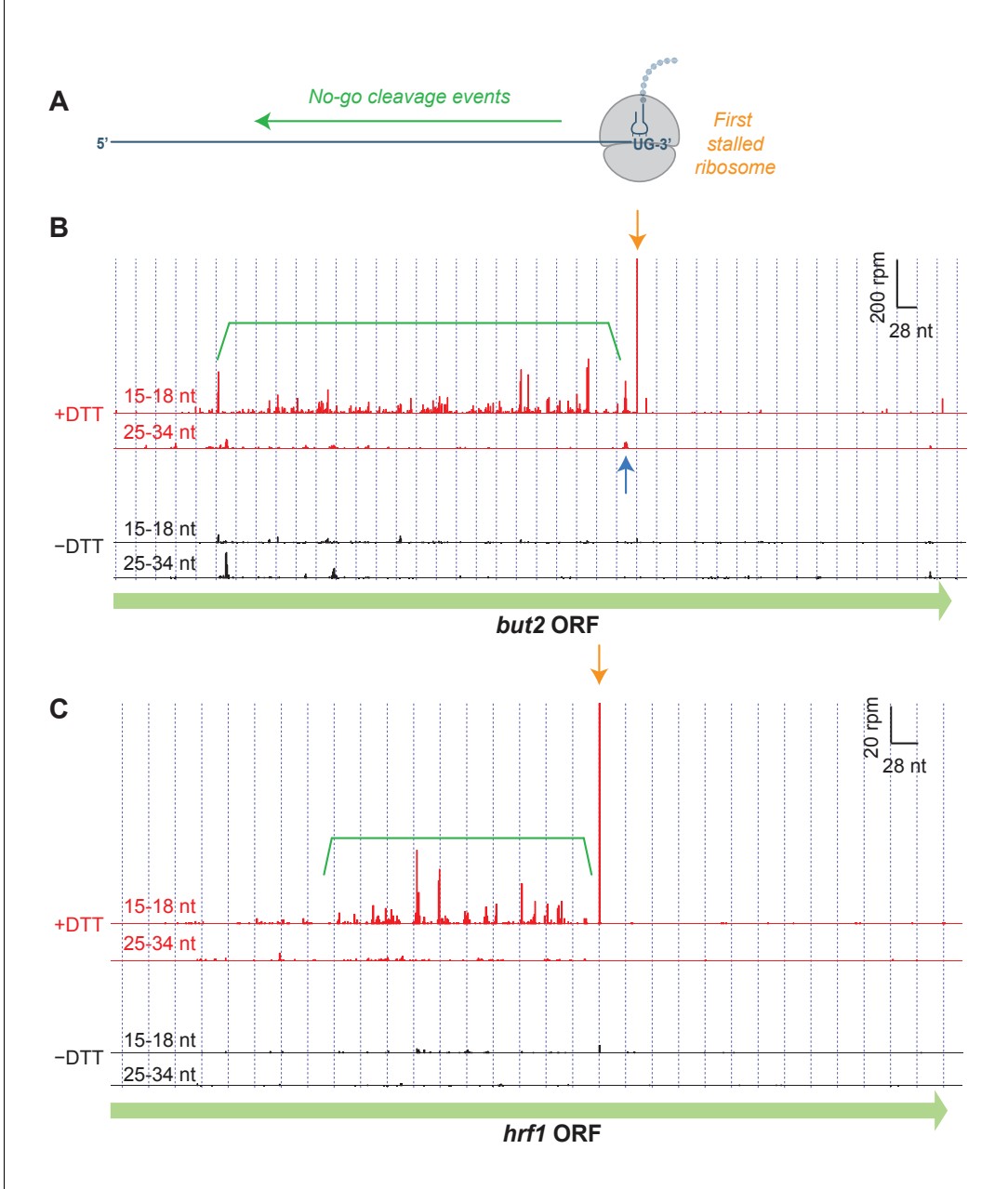

**Figure 3.** Short read alignments to individual example genes reveal no-go decay. (**A**) Model for how Ire1 cleavage leads to ribosome stalling when the 3′ UG is positioned in the A site. This stalled ribosome triggers upstream cleavage of the mRNA (no-go decay, green arrow). These upstream cleavages are expected to, in turn, stall additional ribosomes. (**B**) Example of (3′ end assignment) ribosome profiling data from the *dom34Δ/ski2Δ* strain on the gene *but2*. The data show an initial Ire1-mediated cleavage at a UG/C site (orange arrow). Upstream cleavage events (no-go decay) are evident as peaks in the short-footprint data (ribosomes stalled on truncated mRNA ends) in the presence of DTT (green bracket). The peak in the long-read data (blue arrow) suggests a 'disome' structure forms when an upstream ribosome runs into the ribosome stalled on the 3′ terminal UG. (**C**) Same as (**B**) but for the gene *hrf1*.

DOI: https://doi.org/10.7554/eLife.29216.007

The following figure supplements are available for figure 3:

**Figure supplement 1.** Footprint data (3′ end assignment) from the *dom34Δ/ski2Δ* strain on the gene *ire1*.
DOI: https://doi.org/10.7554/eLife.29216.008

**Figure supplement 2.** Read alignment to an example gene in different strain backgrounds.
DOI: https://doi.org/10.7554/eLife.29216.009

**Figure supplement 3.** Read alignment to an example gene in different strain backgrounds.

*Figure 3 continued on next page*

*Figure 3 continued*

DOI: https://doi.org/10.7554/eLife.29216.010

(*Guydosh and Green, 2017*). Moreover, aligning the short (15–18 nt) ribosome reads by 3' their ends allowed us to pinpoint the site of mRNA cleavage that caused ribosome stalling (*Figure 3A*). As we show in the example of *but2* mRNA in *Figure 3B* (orange arrow), after UPR induction in *dom34Δ/ski2Δ* cells we typically observed a striking enrichment in short (15–18 nt) reads at a UG/C site (where cleavage occurs between the G and C), consistent with Ire1's known RNA substrate specificity (*Gonzalez et al., 1999*). In some cases, we also saw enrichment of long (25–34 nt) footprints with a 3' end positioned ~15 nucleotides (roughly half of a long footprint) upstream of the short reads (*Figure 3B*, blue arrow). The relative positioning of these reads suggests the generation of a 'disome' (two stacked ribosomes) at this UG/C site. The significance of this observation is discussed further below.

We also noted a strong UPR-dependent enrichment of short reads that extended typically hundreds of nucleotides on the 5' side of the UG/C site. These upstream footprints are consistent with extensive mRNA cleavage events upstream of the initiating cleavage. These observations are reminiscent of NGD patterns (*Figure 3A*) that have previously been documented in *S. cerevisiae* (*Guydosh and Green, 2014*; *2017*). Our data here demonstrate that this process is directional and extends over long distances from precise stall sites. In many examples (*Figure 3C*), the pattern of cleavage showed periodicity, suggesting that iteratively stalled ribosomes in this background play a critical role in determining the no-go cleavage pattern (further analyzed below). Intriguingly, we found that one of the targeted transcripts is that encoding Ire1 itself (*Figure 3—figure supplement 1*). We also examined the effects in WT strains or where only *ski2* or *dom34* was individually knocked out (*Figure 3—figure supplement 2*, *Figure 3—figure supplement 3*). We saw little or no evidence for stalling or NGD patterns in WT and *ski2Δ* strains. However, in the *dom34Δ* strain, we observed evidence of the NGD pattern in both examples and for stalling in one case (*Figure 3—figure supplement 2*).

## Short read accumulation reveals an expanded list of Ire1 mRNA targets

To more precisely pinpoint the sites of ribosome stalling and to identify the full scope of RIDD targets, we next examined the effect of UPR induction on short (15–18 nt) footprint density at every individual nucleotide position. Given the large size of the *S. pombe* genome (>10 million single base positions), we initially focused on nucleotide positions in genes with the most reads (>1.5 rpm in both, i.e.,±UPR induction samples) for data visualization (*Figure 4A*). As expected, read density between UPR-induced and uninduced samples was correlated because many factors (independent of UPR induction) determine the pattern of read density within a gene. However, we noted a strong (>10 x) enrichment in read density for a distinct cluster of UG/C sites upon UPR induction (*Figure 4A*, left panel; region above green diagonal line). Compared with cleavages at non-UG/C sites that also met this criterion (*Figure 4A*, center panel; region above green diagonal line), UG/C sites were notably overrepresented (7.9% of UG/C sites vs. 0.0021% of non-UG/C sites; note that histogram is on a log scale), suggesting that they represent primary cleavage sites of Ire1. Moreover, the Ire1 cleavage sites previously identified by 2', 3'-cyclic phosphate sequencing (*Kimmig et al., 2012*) that met the 1.5 rpm threshold were enriched in the UPR-induced sample, most more than 10-fold (*Figure 4A*, right panel).

Further analysis of the full set of data mapping to the transcriptome (without the minimal read threshold and including sites without any reads in the case where the UPR was not induced) revealed 5294 sites with >10 fold enrichment under UPR induction and >2 fold enrichment above the background level in their respective gene (*Supplementary file 2*). We found that ~22% of these corresponded to UG/C motifs (*Figure 4B*, blue sector). Mapping these reads to individual transcripts revealed 1287 affected mRNAs, encompassing about a quarter of the *S. pombe* transcriptome. In particular, 471 mRNAs included reads with at least one UG/C motif, and, of these, 91% are predicted to be associated with the ER because of the presence of a signal sequence or transmembrane domain in the encoded protein (*Figure 4B*, purple bar). This high prevalence of ER association is consistent with these genes representing a list of RIDD targets, expanded well beyond that reported

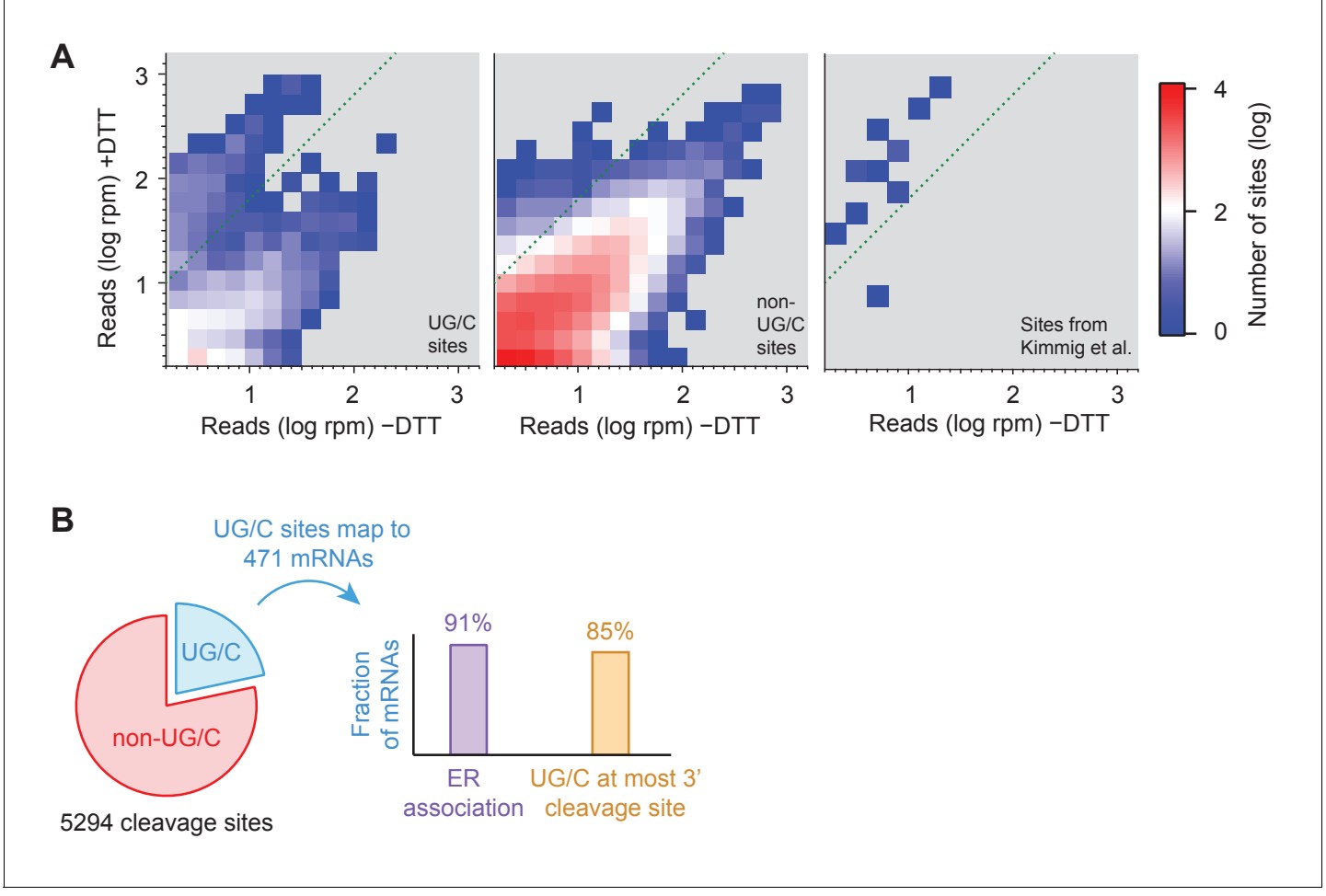

**Figure 4.** Identification of novel mRNA cleavage sites generated by the UPR. (**A**) Histograms of short-read footprint density at individual nucleotide positions across the transcriptome in conditions of +DTT and −DTT in the *dom34Δ/ski2Δ* strain. Counts above dotted line represent loci where reads enrich >10 fold when DTT was added, the threshold used, in part, to identify Ire1 cleavage sites in all downstream analysis. Data for UG/C and non-UG/C sites are separated (left and center panels, respectively), revealing that stalled ribosomes tend to enrich at UG/C sites more than non-UG/C sites (more counts above green line). In addition, UG/C sites from 38 cleavage sites (corresponding to 23 mRNAs) previously identified (*Kimmig et al., 2012*) to be Ire1 targets by 2', 3'-cyclic phosphate sequencing are shown (provided they meet a 1.5 rpm minimal threshold) for reference (right panel) and serve as a positive control for our method. (**B**) Breakdown of all identified cleavage sites by nucleotide motif (pie chart) and analysis of the 471 transcripts to which the UG/C sites map (right). Most (91%) mRNAs with a UG/C cleavage site are associated with the endoplasmic reticulum, as expected for Ire1 targets (purple bar). In addition, 85% of the most downstream cleavage sites on these transcripts corresponded to a UG/C site, consistent with the proposed no-go decay mechanism (orange bar).

DOI: https://doi.org/10.7554/eLife.29216.011

The following figure supplement is available for figure 4:

**Figure supplement 1.** Transcriptome mapping of genes with cleavage sites.
DOI: https://doi.org/10.7554/eLife.29216.012

previously. Consistent with this interpretation, we found these 471 mRNAs were clustered similarly to the previously-reported targets when we examined changes in overall levels of 15–18 nt footprints or mRNA-Seq (*Figure 4—figure supplement 1A*). Similar clustering was also evident in mRNA-Seq data where *ire1* was knocked out (*Figure 4—figure supplement 1B*).

In addition, the most downstream cleavage site in this subset of 471 mRNAs occurred at a UG/C about 85% of the time (398 genes; *Figure 4B*, orange bar), consistent with the cleavage patterns described above (*Figure 3A,B*), where a UG/C cleavage event appears to trigger upstream no-go decay at non-UG/C sites. Of the remaining 816 transcripts where no cleavage sites mapped to UG/C

sequences, about 24% of the mRNAs included at least one cleavage-site motif differing from UG/C by only a single base, suggesting that Ire1 may tolerate imperfect recognition signals.

## Frame analysis reveals specificity of cleavage and decoding activity of ribosome on truncated mRNAs

We next binned UPR-dependent strong stall sites (>100 x the background level in a gene) at UG/C motifs into three groups according to whether the terminal G was found in frame 0, frame 1, or frame 2 (*Figure 5A*). Because the ribosome maintains the reading frame during translation and because the 5' end of the footprint is trimmed flush against the ribosome by RNase 1 during preparation of sequencing libraries (*Guydosh and Green, 2017*), we expected that the length of a fragment should depend on the reading frame occupied by the terminal G. We found this to be true: the 16 nucleotide-long short reads were predominantly in frame 0, the 17 nucleotide-long ones in frame 1, and the 18 nucleotide-long ones in frame 2. These observations suggest that the ribosome halts when the 3' end of the mRNA is positioned randomly in the A site, consistent with the idea that successful decoding requires that the A site is filled with an intact codon. We also noticed a minority population of 15-nt reads for UG/C motifs in frame 2 (*Figure 5A*, left tail of orange curve). The existence of this population suggests that the ribosome can decode the mRNA when the A site is filled with the terminal 3' nucleotides of a truncated mRNA, which at a low efficiency of ~1/3 of the time can translocate yielding the 15 nucleotide-long footprint.

Examination of ribosome footprint positions at strong stalls showed that the distribution of Ire1 UG/C cleavage sites across the 3 reading frames followed the underlying bias in the transcriptome (*Figure 5B*, compare top and bottom). Analysis of the UG/C target through the MEME algorithm (*Moreno et al., 1991*) did not reveal any further strong features for the documented cleavage events. This analysis suggests that Ire1 is not influenced by recognizable sequence context immediately outside of the UG/C motif (*Figure 5C*).

By contrast to the UG/C sites, we found that the non-UG/C sites harbored some preference for a particular reading frame (note that frame 0 non-UG/C cleavages are diminished relative to the background frequency, *Figure 5B*). Because many of these non-UG/C cleavage events likely result from NGD, we might expect that the (in-frame) stalled ribosomes that trigger this process guide the NGDase, which could account for this bias. This finding in fission yeast is consistent with our prior work in budding yeast that also showed a frame preference for no-go cleavage events that take place upstream of ribosomes stalled in poly(A) tails (*Guydosh and Green, 2017*).

## Trends in average pause heights reveal importance of both no-go (Dom34) and mRNA decay (Ski2) pathways

To analyze the pattern of NGD cleavage events upstream of primary Ire1 cleavages (*Figure 3*) in further depth, we aligned 107 sequences that triggered the strongest ribosome stalling events (>200 x above gene background level and >10 rpm upon UPR-induction) at UG/C motifs in frame 0 and overlayed the short (15–18 nt) ribosome profiling data from WT and different mutant strains (*Figure 6A*, note that * symbols denote some peaks are scaled). In these averages, the strong peak centered at '0' (the UG/C site used as anchor in the alignment) upon UPR induction ('+DTT') corresponds to a ribosome stalled at the site of Ire1 cleavage. This peak was strongest in the UPR-induced *dom34Δ/ski2Δ* strain (*Figure 6A*, red traces) but is also evident, albeit to a lesser extent, in the UPR-induced *dom34Δ* and *ski2Δ* strains (*Figure 6A*, green and blue traces, respectively). Its dependence on UPR induction established that the Ire1 cleavage event is activated by ER stress and its dependence on the elimination of Dom34 and/or Ski2 establishes its dependence on the ribosome rescue and the 3'→5' decay pathways. The variance of the peak heights is consistent with the growth data in *Figure 1* showing that deletion of either *dom34* or *ski2* increased the mutant cells' sensitivity to ER stress. The observation of a small peak at position 0 in the *dom34Δ/ski2Δ* strain in the absence ER stress (*Figure 6A*, red traces) suggests that Ire1 becomes partially activated under these conditions.

These findings are further supported by analysis of the size of ribosome footprints that map to strong cleavage sites in frame 0 (pause score >100 x background level upon UPR induction). The relative proportion of short reads (e.g. 16 nucleotide reads) increased upon *dom34* deletion or UPR

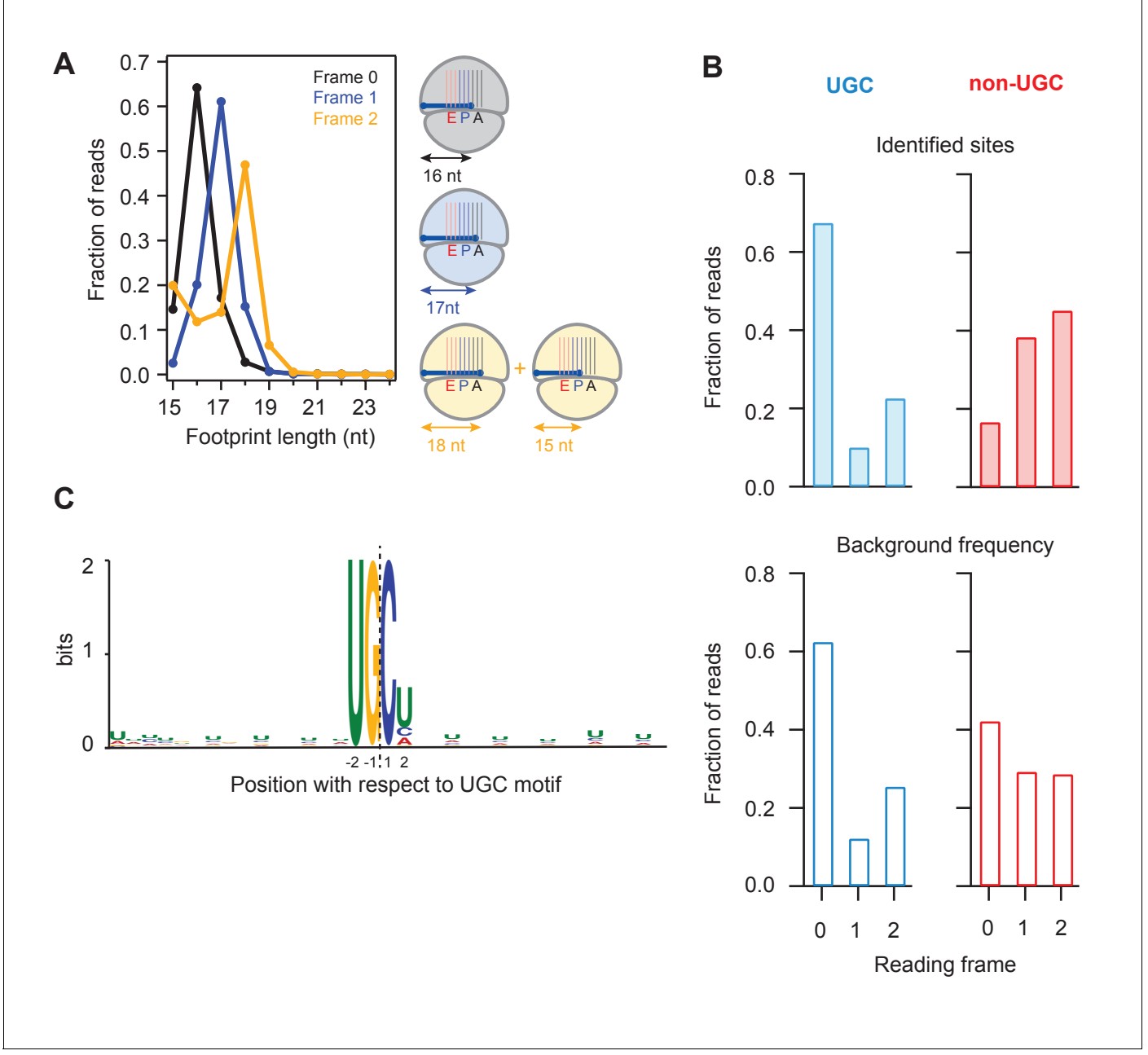

**Figure 5.** Analysis of frame and sequence context of cleavage sites. (**A**) Analysis of footprint sizes from reads that map to the library of 24-nt sequences immediately upstream of strong UG/C cleavage sites (pause score >100 in the presence of DTT) found in frames 0, 1, or 2. The peak position changes according to frame because the ribosome, which protects the footprints, moves in 3-nt increments. When UG/C is in frame 2 (all 3 nucleotides in the A site and footprint size measures 18 nt), about 1/3 of ribosomes manage to move forward, positioning UG/C in the P site and shortening the footprint to 15 nt. From previous work, we know that the A site of the ribosome lies 16–18 nt from the 5' end of the footprint (*Ingolia et al., 2009*). (**B**) Reading frame of the terminal base for strong UG/C cleavage sites (blue, due to Ire1) and non-UG/C sites (red, due to mostly to no-go decay) identified with pause score >100 with DTT present (top). Background reading frame frequency of these motifs in the transcriptome is shown for reference (bottom). (**C**) Motif analysis using MEME (*Bailey et al., 2009*), of all identified UG/C cleavage sites in frame 0 shows little outside sequence context.

DOI: https://doi.org/10.7554/eLife.29216.013

induction (*Figure 6B*) with respect to long (~28 nucleotide) reads. This observation indicates that most ribosomes found at these sequences are stalled.

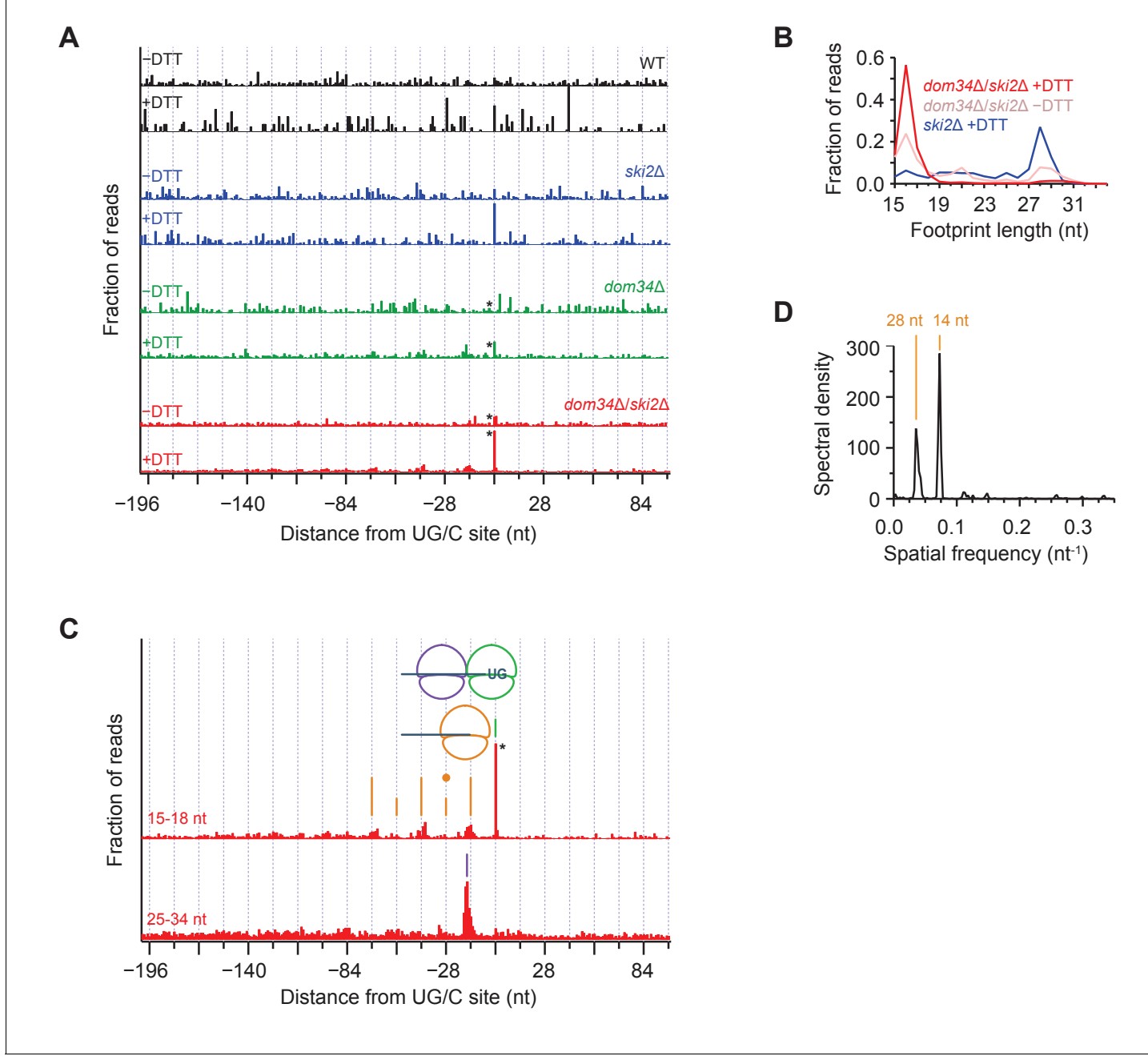

**Figure 6.** Average ribosome positioning during no-go decay across strain backgrounds. (**A**) Average short ribosome footprint density (3' end assignment) at identified UG/C sites in frame 0 for multiple strains. Only cleavage sites with pause scores > 200 in presence of DTT and at least 10 rpm of read density in presence of DTT are included to improve resolution. Analysis of data in frame 1 or frame 2 separately appeared similar but was not included here because the variation in footprint size by frame (**Figure 5A**) tends to blur the peaks. Note that * indicates peak height reduced 10x for space constraints. Knockout of *dom34* and *ski2*, as well as addition of DTT, all enhance observation of ribosome stalling at UG/C sites (position 0). (**B**) Size analysis of short footprints that map to the 34-nt region immediately upstream of strong UG/C sites (pause score >100) in frame 0 shows that the 15–18 nt reads are preferred over background reads more strongly in the presence of DTT and absence of *dom34*. (**C**) Same as (**A**) for *dom34Δ/ski2Δ* in the presence of DTT for 15–18 nt and 25–34 nt footprints. Stalled ribosome peak at initial Ire1 cleavage site is indicated (green line). Upstream peaks at 14-nt intervals show evidence of ribosome stalling at no-go decay cleavage events (orange lines). Density at alternating 14-nt intervals is reduced (short orange lines) due to formation of a disome that protects the mRNA. In particular, the first short orange line upstream of the UG/C site (marked with orange dot) is protected by the disome shown in the cartoon. Direct evidence of disome formation is visible as a peak in the 25–34 nt footprint data located ~16 nt upstream of the UG/C cleavage site (purple line). Note that * indicates peak height was reduced 10x for space constraints. (**D**) Power spectrum of the autocorrelation of data in the region 200 nt upstream of the short footprint data in (**C**). Peaks at 14 nt and 28 nt are consistent with the

*Figure 6 continued on next page*

*Figure 6 continued*

alternating 14-nt intensity shown in (C). The stronger amplitude of the 14-nt peak reveals that majority of cleavage pattern is due to monosome formation. The smaller peak at 28 nt shows the contribution of disomes.

DOI: https://doi.org/10.7554/eLife.29216.014

The following figure supplement is available for figure 6:

**Figure supplement 1.** Same as *Figure 6C* but for all identified cleavage sites.

DOI: https://doi.org/10.7554/eLife.29216.015

## Periodicity of ribosome footprints suggests that a NGDase tightly associates with ribosomes

We next focused on the short (15–18 nt) footprint dataset in the *dom34Δ/ski2Δ* strain along with the corresponding long (25–34 nt) footprints (*Figure 6C*). When we turned our analysis to the regions upstream of the UG/C target sites, we noticed a striking periodicity in the 15–18 nt dataset: peaks repeated roughly every 14 nt, decreasing in abundance as one moves upstream of the initial cleavage site (*Figure 6C*, orange lines). To confirm the accuracy of this distance measurement, we computed the power spectrum of the autocorrelation of the repeat region (*Figure 6D*), which revealed a strong correlation every 14 nucleotides and, to a lesser extent, every 28 nucleotides (discussed further below). The regularity in spacing of these peaks can be accounted for by a model wherein a ribosome that is stalled at a UG/C motif initiates NGD through endonucleolytic cleavage immediately upstream of it. This cleavage event, in turn, stalls the next ribosome behind it, generating another peak just 14-nucleotides upstream. In this way, the ribosome serves as a ruler that templates the repeat pattern.

## Disomes form at sites of Ire1 cleavage

As suggested by the power spectrum peak at 28 nucleotides, the 14-nucleotide repeat pattern appeared to be superimposed by a 28-nucleotide repeat pattern, resulting in peaks with a higher amplitude at alternating 14-nucleotide positions (*Figure 6C*, alternating height of orange lines). This trend is also visible in the data upstream of all UG/C cleavage sites (908 total sites, including both strong and weak stalls) (*Figure 6—figure supplement 1*). A simple explanation for this observation is that two ribosomes occasionally stack — forming 'disomes' — when a tail-gating translating ribosome rear-ends a ribosome stalled at a mRNA truncation site before the NGDase cleaves the mRNA. This notion is supported by the analyses of the averaged dataset of the long, 25–34 nucleotide footprint data that reveal a dominating peak positioned precisely where we expect two ribosomes to collide (*Figure 6C*, purple line). In this scenario, the upstream ribosome in the disome protects the mRNA from cleavage at the site of the 2nd NGD site (*Figure 6*, orange dot), thereby accounting for the reduced amplitude in arrested short-footprint ribosomes at that position. The evidence for a continued (though diminishing) pattern of alternating strong and weak peaks further upstream suggests that higher-order ribosome structures (i.e. mass-collisions leading to trisome, quadrasome, etc.) can form at a UG/C site. Alternatively, disome formation may be followed by cleavage, creating a new end on which the process repeats, leading to successive disome formation at upstream sites of no-go cleavage.

## Discussion

### Molecular model for Ire1-triggered NGD during ER stress in *S. pombe*

We provide strong support for the coupling of NGD with Ire1-triggered endonucleolytic mRNA cleavage during the UPR in *S. pombe*. Our observations suggest a model (*Figure 7*) wherein initial Ire1-triggered cleavage at UG/C sites in ER targeted mRNAs results in ribosome arrest at the truncated end of the mRNA, generating short footprints in ribosome profiling experiments. Ribosomes stalled at the end of the truncated mRNAs can trigger stalling of trailing ribosomes by a 'fender-bender' type collision, causing them to stack as disomes and perhaps larger stacks. Single ribosomes and ribosome stacks trigger the endonucleolytic cleavage events of the NGD pathway. Our model posits that ribosome stalling and mRNA cleavage is then reiterated until the upstream mRNA is

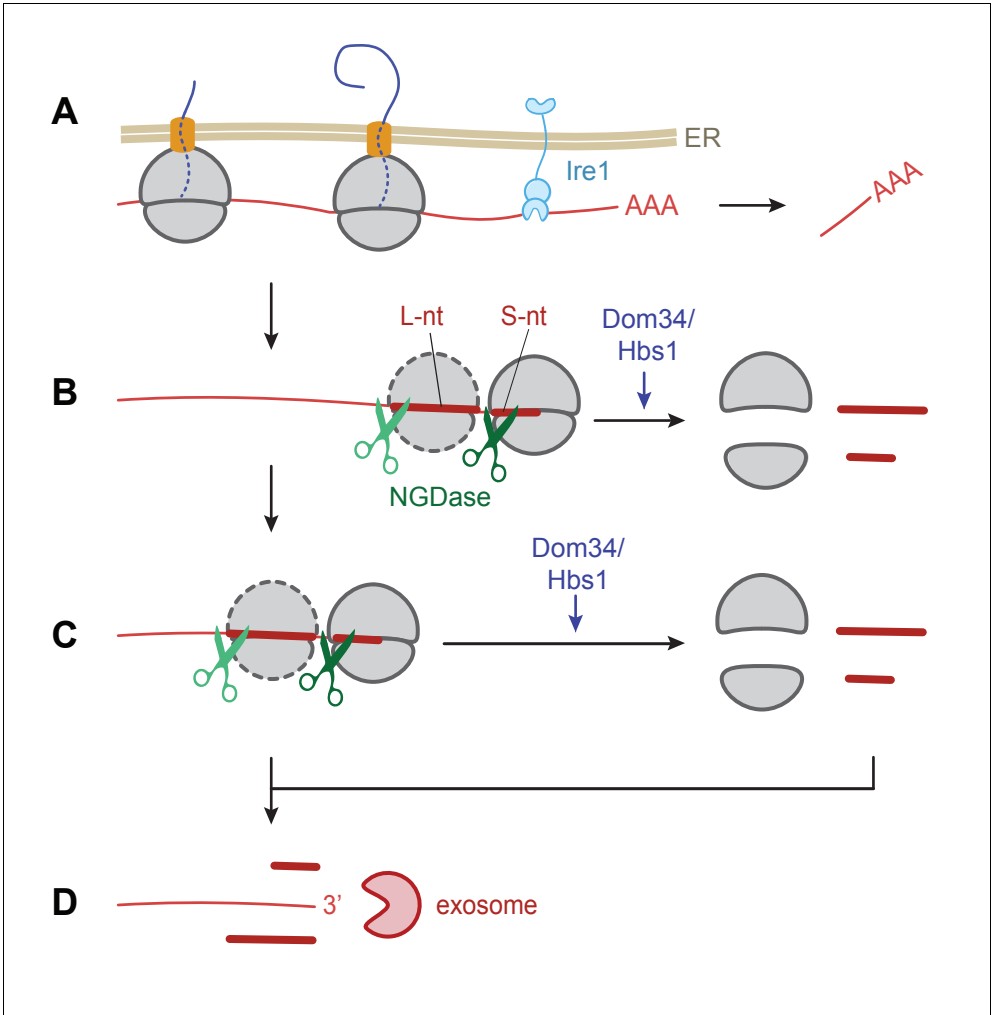

**Figure 7.** Model of Ire1 mRNA cleavage of leading to no-go decay. (**A**) Initial Ire1 cleavage of a target mRNA at UG/C sites within the open reading produces a truncated mRNA lacking its poly(A) tail. (**B**) Ribosomes continue to elongate until they reach the end of mRNA fragment and stall with an empty A-site. The stalled ribosome recruits NGDase, which cleaves the mRNA immediately upstream of the ribosome as part of the no-go decay process. Dom34/Hbs1 then recycle the ribosome. In their absence, ribosomes protect a short nucleotide ('S-nt'), which we identify experimentally. If NGDase cleavage is slow enough to allow another ribosome to stack onto the leading one, a disome is formed. The trailing ribosome protects a long nucleotide ('L-nt'). The tendency to form disomes (or perhaps even larger stacks) in vivo will depend on the relative kinetics governing elongation, NGDase cleavage, and ribosome recycling. (**C**) The process can repeat as more ribosomes arrive at newly generated 3' ends. (**D**) Finally, the exosome degrades the mRNA fragments and any intact piece of 5' mRNA that do not have a stalled ribosome protecting their 3' end. For clarity, the ER membrane and nascent peptides are not shown in panels **B–D**.

DOI: https://doi.org/10.7554/eLife.29216.016

degraded into short ribosome-associated fragments and longer ribosome-free 5' mRNA fragments subject to unobstructed 3'→5' mRNA decay by the cytosolic exosome.

Whether an Ire1-targeted transcript is fully sliced into small fragments by NGD or whether the exosome manages to degrade unobstructed regions of some transcripts before ribosome stalling and NGD cleavage takes place likely depends on the relative kinetics of the NGDase, exosome, rescue activity of Dom34/Hbs1, speed of elongation by the ribosome, and ribosome loading (translational efficiency) of the transcript. While the clearest evidence for enrichment of periodic short footprint ribosomes emerges in the *dom34Δ/ski2Δ* mutant cells, weaker trends are seen in either single mutant alone (*Figure 6A*), suggesting that both the NGD/ribosome rescue pathway (Dom34/

Hbs1/NGDase) and the 3′→5′ mRNA decay pathway (exosome/SkiX) are critical for the elimination of the Ire1 cleavage products in WT cells. It is striking that the effects of either single knockout on ribosome stalling and upstream NGD were small compared to the effects of the double knockout. These observations suggest that the two pathways can partially substitute for each other.

This finding is bolstered by the phenotypic assay showing loss of either pathway results in less tolerance for ER stress (*Figure 1*). The assay, in particular, showed that both Ski2, a component of the Ski complex, and Ski7, are involved. While Ski7 interacts with the Ski complex and the RNA exosome, it has been suggested to have additional effects on translation and to bolster tolerance to stress (*Jamar et al., 2017*; *Kowalinski et al., 2016*). Ski7 would therefore be an interesting candidate for further mechanistic studies.

## Implications to the general mechanism of NGD

Unlike previous studies of NGD at less-well defined ribosome stall sites (i.e. poly(A) stretches or long stem-loops), the endonucleolytic cleavage sites detected here were precise and thus offer the opportunity to more clearly examine the mechanism of long-distance NGD-cleavage events. Based on the dimension of the ribosome and positional registration of the mRNA footprint gleaned from previous ribosome profiling studies, we can infer mechanistic details regarding the still unknown and highly sought-after NGDase. For example, in the analysis of the repetitive pattern of protected fragments upstream of the primary UG/C cleavage site, we found that peaks were separated by only 14 nucleotides, which is different from a distance corresponding to the 15–18 nucleotide range observed in the footprint distributions created after RNase 1 treatment (*Figure 5A*). This observation implies that the NGDase does not cleave flush against the upstream face of the ribosome, as occurs during experimental preparation of the 15–18 nt footprints. Instead, these data suggest that NGDase cleavage must occur *inside* the ribosome, at a position that is found one or two nucleotides within the channel from which the mRNA emerges the ribosome. It is also possible that the structure of ribosomes stalled in this way may be more flexible and therefore allow NGDase access to this site without having to reach inside the ribosome. This conclusion is consistent with our findings that NGD cleavage events show frame bias (*Figure 5B* and previously in *S. cerevisiae* [*Guydosh and Green, 2017*]), reinforcing the notion that the no-go cleavage events take place on the ribosome.

## The Ire1 RIDD pathway collaborates with NGD to maintain ER homeostasis

By identifying ribosomes stalled on mRNAs that are cleaved by Ire1 upon ER stress, our data reveal that the scope of RIDD upon UPR induction is far broader than appreciated to date. By comparison to the cohort of 39 RIDD target mRNAs identified previously by mRNA-Seq (*Kimmig et al., 2012*), we here identified 471 mRNAs whose degradation is induced by Ire1-mediated cleavage at UG/C sites. This group of 471 mRNAs includes 34 of the 39 previously identified mRNAs (three could not be evaluated due to little or no read depth; the other two were successfully identified by cleavage at non-UG/C sites), thus expanding the set of Ire1 substrates by over ten-fold. Almost all (91%) of the identified mRNAs encode proteins that bear an ER-directed signal sequence or transmembrane domain and thus are predicted to be translated on the ER membrane in which Ire1 resides. The target list contains many mRNAs encoding proteins with functions in the secretory pathway and lipid metabolism, indicating that the regulation of these proteins' biosynthesis may serve to fine-tune ER homeostasis, perhaps by adjusting the lipid composition of the membrane (*Volmer and Ron, 2015*). We also found that Ire1 appeared to cleave its own mRNA, suggesting a potential autoregulatory mechanism to limit production of this endonuclease once a threshold level is reached. We note that our method of detecting mRNA cleavage via the presence of a stalled ribosome limits our ability to detect cleavages outside canonical coding regions. It has previously been shown that Ire1 can target the 3′UTR (*Kimmig et al., 2012*), and it is therefore reasonable to assume that additional cleavage sites may be found in the 5′UTR or 3′UTR regions.

We also identified 816 mRNAs that are cleaved upon UPR induction, even though the cleavage sites did not correspond to UG/C motifs. While 193 of these mRNAs include sites matching UG/C with only a single base change and are therefore putative targets of Ire1, the remaining 623 mRNAs exhibit ribosome stalling solely at other classes of endonucleolytic cleavage sites that are induced upon UPR induction. One possible explanation is that Ire1 or another, yet to be identified

endonuclease that is activated by ER stress, can cleave at these other sites. While UG/C sites were strongly enriched in the set of Ire1 target mRNAs, the vast majority of UG/C motifs that we evaluated did not meet our threshold criteria for inclusion: reads representing only 7.9% of the mRNAs containing UG/C motifs shown in *Figure 4A* (left panel) were enriched >10 x (i.e., fell above the green line) when the UPR was induced. It is possible that this number could be increased with improved methodology for cleavage site detection; yet it seems more plausible that other features of these sites are critical in determining the efficiency of Ire1 target selection. From the data shown in *Figure 5C*, it seems unlikely that such features lie in the immediate sequence context of the site. Other possibilities that could offer an explanation include higher-order mRNA structural features, mRNA localization to the ER, a specific complement of RNA binding proteins, or features in the nascent polypeptide chains. Since it is clear that ER-associated factors are enriched in the set of targeted mRNAs, we asked whether the ER-associated mRNAs that manage to escape cleavage showed any particular functional enrichment. We were unable to reveal any trend, further implying that additional properties of these transcripts are involved in specifying targeting.

The biological importance of this broadened spectrum of Ire1 RIDD targets is underscored by the genetic screen that identified components in ribosome rescue and nonstop decay in an unbiased fashion. In particular, mutants in which the Ski-complex was defective showed strong sensitivity to ER stress. One potential explanation for why the failure to clear truncated mRNAs at the ER membrane may be so severe is that the stalled ribosomes may clog translocons and limit protein flux into the ER (*Arakawa et al., 2016*), including that of newly synthesized chaperones and other factors required to restore ER homeostasis. If this were the case, we expect these findings will apply to higher eukaryotes where the RIDD pathway is also active and, as such, have broad implications for human disease. The UPR triggers the integrated stress response (ISR), and many recent reports have suggested that chronic ISR activation by unfolded proteins or other stresses can lead to a number of diseases, including atherosclerosis (*Tufanli et al., 2017*) and many forms of neurological dysfunction (*Scheper and Hoozemans, 2015*). The ribosome recycling and mRNA decay pathways that we have shown here to be intricately intertwined are likely to be important for maintaining fitness of the proteome and human health.

## GEO accession codes

All high throughput data have been deposited with NCBI GEO with accession number GSE98934.

## Materials and methods

### Strain creation

Standard cloning and yeast techniques were used for construction, transformation and gene deletions as described previously (*Moreno et al., 1991*). Strains used in this study are listed in *Supplementary file 3*. All non-ribosome profiling experiments were carried out in yeast extract complex media (YE5S) supplemented with 0.225 mg/ml of l-histidine, l-leucine, l-lysine, adenine and uracil at 30°C, unless otherwise described.

### Chemical genomic screen

The *Schizosaccharomyces pombe* Haploid Deletion Mutant Set version 2.0 (M-2030H; Bioneer Corporation) was accessible through the Azzalin lab (ETHZ). The original library contained 3006 non-essential gene deletions, but only 2346 non-essential gene deletions were viable in this study. The library was spotted in duplicates on a 384-array format with YE5S media supplemented with or without 0.15 µg/ml tunicamycin. Plates were incubated at 30°C and after 3 days pictures were taken by the Fusion solo S system. Colony sizes were quantified and analyzed with Balony software (https://code.google.com/p/balony/). Resistant and sensitive gene hits were identified by the described z-score threshold. Growth rates for each deletion strain are listed in *Supplementary file 1*.

### Ribosome profiling

All cells were grown in YES 225 media (Sunrise bioscience). All media was sterile filtered and cultures were grown at 30°C. Cultures were harvested at an OD of ~0.6 after~5 doubling times. DTT was added to 1 mM at 1 hr prior to harvest.

Ribosome profiling libraries were prepared as described (*Guydosh and Green, 2014*) by using a protocol very similar that used by Ingolia and coworkers (*Ingolia et al., 2012*). All RNA size separation gels were cut as a single slice from 15 to 34 nt for footprints and 40–60 nt for mRNA-Seq. All footprint samples were lysed and separated over sucrose gradients in the presence of 0.1 mg/ml CHX. Total mRNA for mRNA-Seq was isolated from cells using hot SDS/acid phenol and chloroform, as previously described. Footprint samples here were subject to rRNA subtraction by using a yeast Ribo-Zero kit (Epicentre). Subtraction of rRNA for all footprint samples was performed prior to linker ligation with the exception of the *dom34Δ/ski2Δ* strains where it was performed after linker ligation. This change was implemented because the RiboZero kit introduces a variety of short sequences that map at random across the genome, leading to occasional spikes in the data (*Guydosh and Green, 2017*). Samples for mRNA-Seq were subject to subtraction after purification of total RNA (mRNA-Seq). The 50°C incubation step for standard footprint preparation was skipped in the Ribo-Zero-modified protocol, as recommended by the manufacturer. Sequencing and demultiplexing were performed on an Illumina HiSeq2500 at the Johns Hopkins Institute of Genetic Medicine.

## Deep sequencing analysis

Analysis of footprints was essentially as described (*Guydosh and Green, 2014*) with modifications as noted below. The PomBase ASM294 v2.22 genome assembly was the reference genome used for analysis (*Wood et al., 2012*). De-multiplexed sequences were processed to remove reads with any position with Phred score <20 or assigned N as a quality filter step. Following a search for the linker and sorting of reads into short (15–18 nt) or long (25–34 nt) populations, contaminating ladder oligonucleotides were removed and alignment to a database of rRNA and tRNA spliced genes was performed. Following this step, a second round of subtraction for short, 15–18 nt, reads was performed by aligning to all the tRNA gene sequences plus extension with CCA on their 3′ ends. This enhanced the removal of cytoplasmic tRNA fragments. The remaining reads were mapped to the genome and those that failed to match were aligned to a custom transcriptome, created by splicing together annotated exons. Read lengths were assessed with the FastQC software (Babraham Bioinformatics).

All reads that aligned to multiple coding sequences were discarded. Read occupancy was determined by giving one count per read at its 3′ end and in some cases shifted to align with various active sites in the ribosome (i.e. start of the P site) as described below. However, reads were assigned to 5′ ends for mRNA-Seq analysis. These mRNA-Seq reads were also trimmed of 3′ consecutive As after alignment and remapped to include those near poly(A) tails, as was done previously (*Guydosh and Green, 2014*). Read counts were then normalized by dividing by the total number of million mapped reads in a sample. Alignments were performed with Bowtie (*Langmead et al., 2009*) and included the parameters: -y -a -m 1 –best –strata. All footprint alignments to coding sequences allowed for no mismatches; mRNA-Seq alignments allowed two mismatches. All other analysis software was custom coded in the Python 2.7 programming language and Biopython (*Supplementary file 4*). Plot construction and correlation analysis was done with Igor Pro (Wavemetrics). In general, regions of transcript analysis that overlapped with other transcripts on the same strand or marked dubious were ignored in the analysis.

Gene quantitation (shown in *Figure 2B* and others, and used in calculations elsewhere) relied on a shift of −2 for 3′ assignment (short reads) or −14 (long reads) and therefore aligns read density roughly with the P site. Total gene ribosome occupancy was quantitated into density units of reads per kilobase per million mapped reads (rpkm) by taking reads mapping to an annotated sequence and dividing by the gene length in kilobases. The reads from the first and last five amino acids were not included to prevent known artifacts around start and stop codons from skewing results.

Ratio analysis was performed by taking the ratio at every point in the transcriptome between datasets from yeast with and without DTT treatment (*Figure 4A* and *Supplementary file 2*). To be included in *Figure 4A*, >1.5 rpm of density had to be present in both datasets. The threshold for detection under DTT exposure (*Supplementary file 2*) was that the ratio between datasets must be >10 and the size of the peak in the +DTT dataset must be at least 2x higher than the average of reads that map to the gene. There were no read density thresholds in the –DTT dataset but positions without any reads were assigned one read (0.244 rpm) so that a lower limit to the ratio could be computed. For some analyses, these thresholds were raised higher as noted. These ratios were corrected for changes in mRNA levels (using mRNA-Seq) between the two datasets. Position-

average plots were created by averaging together (with equal weight) reads in a defined window for every occurrence of a particular motif (i.e. UG/C) in a list of targets (*Figure 6*).

MEME 4.11.2 was run with these parameters: -dna -oc. -nostatus -time 18000 -maxsize 60000 -mod zoops -nmotifs 5 -minw 34 -maxw 34.

## Acknowledgements
We thank Diego Acosta-Alvear and colleagues for their help and discussion. We also thank the members of the Walter lab and Green lab for critical comments on the manuscript and feedback. We are grateful to Claus Azzlin for providing the *S. pombe* deletion library.

## Additional information

### Competing interests
Rachel Green: Reviewing editor, *eLife*. The other authors declare that no competing interests exist.

### Funding

| Funder | Grant reference number | Author |
| --- | --- | --- |
| Howard Hughes Medical Institute | | Peter Walter<br>Rachel Green |
| National Institute of General Medical Sciences | GM 059425 | Rachel Green |
| National Institutes of Health | | Peter Walter |
| National Institute of Diabetes and Digestive and Kidney Diseases | | Nicholas R Guydosh |
| Human Frontier Science Program | | Phillip Kimmig |

The funders had no role in study design, data collection and interpretation, or the decision to submit the work for publication.

### Author contributions
Nicholas R Guydosh, Conceptualization, Formal analysis, Investigation, Methodology, Writing—original draft, Project administration, Writing—review and editing; Philipp Kimmig, Conceptualization, Data curation, Formal analysis, Investigation, Methodology, Writing—original draft, Project administration, Writing—review and editing; Peter Walter, Conceptualization, Formal analysis, Funding acquisition, Investigation, Writing—original draft, Project administration, Writing—review and editing; Rachel Green, Conceptualization, Formal analysis, Funding acquisition, Writing—original draft, Project administration, Writing—review and editing

### Author ORCIDs
Peter Walter https://orcid.org/0000-0002-6849-708X
Rachel Green https://orcid.org/0000-0001-9337-2003

### Decision letter and Author response
Decision letter https://doi.org/10.7554/eLife.29216.026
Author response https://doi.org/10.7554/eLife.29216.027

## Additional files

### Supplementary files
• Supplementary file 1. Growth of mutants in chemical genomic screen.
DOI: https://doi.org/10.7554/eLife.29216.017

• Supplementary file 2. List of identified Ire1 cleavage sites, including positional read count, pause score, and mRNA-Seq data for the gene.
DOI: https://doi.org/10.7554/eLife.29216.018

• Supplementary file 3. Yeast strains and plasmids used in this study.
DOI: https://doi.org/10.7554/eLife.29216.019

• Supplementary file 4. Source code for custom scripts used in this study.
DOI: https://doi.org/10.7554/eLife.29216.020

• Transparent reporting form
DOI: https://doi.org/10.7554/eLife.29216.021

## Major datasets

The following dataset was generated:

| Author(s) | Year | Dataset title | Dataset URL | Database, license, and accessibility information |
|---|---|---|---|---|
| Guydosh NR, Kimmig P, Walter P, Rachel Green | 2017 | Regulated Ire1-dependent mRNA decay requires no-go mRNA degradation to maintain endoplasmic reticulum homeostasis in *S. pombe* | http://www.ncbi.nlm.nih.gov/geo/query/acc.cgi?acc=GSE98934 | Publicly available at the NCBI Gene Expression Omnibus (accession no: GSE98934) |

The following previously published dataset was used:

| Author(s) | Year | Dataset title | Dataset URL | Database, license, and accessibility information |
|---|---|---|---|---|
| Kimmig P, Diaz M, Lang A, Zheng J, Williams C, Aragón T, Li H, Walter P | 2012 | The unfolded protein response in fission yeast modulates stability of select mRNAs to maintain protein homeostasis | http://www.ncbi.nlm.nih.gov/geo/query/acc.cgi?acc=GSE40298 | Publicly available at the NCBI Gene Expression Omnibus (accession no: GSE40298) |

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
