## [Decision Letter]

Thank you for submitting your article "Regulated Ire1-dependent mRNA decay requires no-go mRNA degradation to maintain endoplasmic reticulum homeostasis in *S. pombe*" for consideration by *eLife*. Your article has been reviewed by three peer reviewers, one of whom, Nahum Sonenberg is a member of our Board of Reviewing Editors and the evaluation has been overseen by Vivek Malhotra as the Senior Editor.

The reviewers have discussed the reviews with one another and the Reviewing Editor has drafted this decision to help you prepare a revised submission.

Summary:

Guydosh and colleagues here show that, in response to endoplasmic reticulum stress, primary Ire1-dependent cleavage of mRNAs encoding secreted proteins lead ribosomes to stall at cleavage points, triggering secondary cleavage through the no-go decay (NGD) pathway. Rapid, IRE1-dependent decay (RIDD) of ER-associated transcripts provides the major unfolded protein response in *S. pombe* and has been observed in animal systems as well. This work provides a comprehensive catalogue of target transcripts, enhanced by genetic inactivation of downstream ribosome-dependent decay pathways. It also demonstrates a major physiological role for cycles of ribosome-dependent cleavage occurring on truncated mRNAs.

The basic phenomenon described by the authors is well supported by ribosome footprint profiling data, and is broadly consistent with previous work on RIDD and NGD. The connection between these two processes, and the discovery of a large class of physiological NGD substrates, will likely attract broad interest.

Essential revisions:

1) The authors show a strong enrichment for transmembrane and secreted proteins among the list of direct RIDD targets, but they don't comment on the converse: are classes of ER-associated mRNAs spared from RIDD, and can we rationalize which transcripts escape cleavage? Likewise, is there a functional enrichment for ER chaperones, etc. among the non-target mRNAs?

2) UG/C sites in the 5' UTR should block initiation and destabilize the mRNA, rather than triggering elongation stalls and progressive NGD. Are there differences between transcripts that contain or lack a UGC before the start of the coding sequence?

3) Likewise, the authors provide a clear and thorough analysis of UG/C cleavages, but these represent numerically a minority of UPR-induced cleavages overall. In the discussion, the authors equivocate between saying that these transcripts are "cleaved by Ire1 upon UPR induction" and pointing out that they could be cut by "another, yet to be identified endonuclease". There's no evidence that these RNAs are direct Ire1 targets, however. Since one class of target transcripts match the defined Ire1 sequence preference well, the possibility of a second, distinct nuclease seems stronger.

4) The study identified 76 genes whose deletion results in the rescue of Tm-induced growth defects (proteins involved in vesicle transport and located on the cell surface), and 106 Tm-sensitizing deletions. The authors focused their study on the latter group, while the first group is uncharacterized. Did the authors perform any validation/analysis of the first group?

5) Ribosome footprinting experiments have been performed on WT, dom34Δ, ski2Δ, and dom34Δ/ski2Δ strains. However, data were not fully presented. Most of the experiments compared WT with dom34Δ/ski2Δ strains. Information about the dom34Δ and ski2Δ strains should be presented.

---

## [Author Response]

1) The authors show a strong enrichment for transmembrane and secreted proteins among the list of direct RIDD targets, but they don't comment on the converse: are classes of ER-associated mRNAs spared from RIDD, and can we rationalize which transcripts escape cleavage? Likewise, is there a functional enrichment for ER chaperones, etc. among the non-target mRNAs?

We agree that this is a reasonable question, particularly because we observed previously that the *bip1* mRNA is an example of a major ER chaperone that is upregulated under protein folding stress. We analyzed the group of non-RIDD targeted mRNAs and the group of the Ire1-cleaved mRNAs against all secretory mRNAs. There was no significant enrichment for secretory proteins in the non-RIDD mRNAs so we, unfortunately, cannot offer a theory to explain which ER-associated mRNAs escape cleavage and why. We have now included the results of this analysis in the Discussion in the paragraph where we discuss target specificity.

2) UG/C sites in the 5' UTR should block initiation and destabilize the mRNA, rather than triggering elongation stalls and progressive NGD. Are there differences between transcripts that contain or lack a UGC before the start of the coding sequence?

We appreciate the thoughtfulness of this question as we had not explicitly considered it. To explore this possibility further, we made a list of all genes with a UG/C in the 5’UTR and checked whether there were notable trends in the gene-level expression of mRNA or footprints. We did not see any pattern emerge.

This is not terribly surprising because mRNA-Seq alone is not very sensitive for identifying cleaved mRNAs. In fact, it only revealed 39 targets in our previous study. Because translation of 5’UTRs is relatively low, our higher-sensitivity short footprint profiling approach is only appropriate for cleavages in heavily translated regions and does not offer a means to detect cleavage events in the 5’UTR. We therefore cannot rule out this possibility and have now added text to the manuscript to include it.

3) Likewise, the authors provide a clear and thorough analysis of UG/C cleavages, but these represent numerically a minority of UPR-induced cleavages overall. In the discussion, the authors equivocate between saying that these transcripts are "cleaved by Ire1 upon UPR induction" and pointing out that they could be cut by "another, yet to be identified endonuclease". There's no evidence that these RNAs are direct Ire1 targets, however. Since one class of target transcripts match the defined Ire1 sequence preference well, the possibility of a second, distinct nuclease seems stronger.

The reviewers make a good point. We’ve adjusted the wording of this paragraph to address this point and improve clarity.

4) The study identified 76 genes whose deletion results in the rescue of Tm-induced growth defects (proteins involved in vesicle transport and located on the cell surface), and 106 Tm-sensitizing deletions. The authors focused their study on the latter group, while the first group is uncharacterized. Did the authors perform any validation/analysis of the first group?

We have not validated in detail the Tm-suppressing deletions but we very much appreciate that these may be interesting. One way to rationalize the enrichment of these gene categories is by the downstream consequences of tunicamycin (inhibition of N-glycosylation). Inhibition of specific parts of this pathway may cause a beneficial reorganization of the secretory system, which lead to a growth advantage for the cell under such circumstances. We now point out this possibility with an added citation in the main text.

5) Ribosome footprinting experiments have been performed on WT, dom34Δ, ski2Δ, and dom34Δ/ski2Δ strains. However, data were not fully presented. Most of the experiments compared WT with dom34Δ/ski2Δ strains. Information about the dom34Δ and ski2Δ strains should be presented.

We appreciate this concern and added a new supplement to Figure 3 to display all the ribosome profiling data that we obtained for the main example genes. We have also added text to the Discussion section to better highlight the analysis that we performed in Figure 6, which specifically examines the effects of the single and double mutations and how these pathways appear to be somewhat redundant under the conditions of our assay.